# Rewired Metabolism Caused by the Oncogenic Deregulation of MYC as an Attractive Therapeutic Target in Cancers

**DOI:** 10.3390/cells12131745

**Published:** 2023-06-29

**Authors:** Laura Vízkeleti, Sándor Spisák

**Affiliations:** 1Department of Bioinformatics, Faculty of Medicine, Semmelweis University, 1094 Budapest, Hungary; vizkeleti.laura@med.semmelweis-univ.hu; 2Institute of Enzymology, Research Centre for Natural Sciences, Eötvös Loránd Research Network, 1117 Budapest, Hungary

**Keywords:** protooncogene, MYC deregulation, altered cellular metabolism, therapeutic targets

## Abstract

MYC is one of the most deregulated oncogenes on multiple levels in cancer. As a node transcription factor, MYC plays a diverse regulatory role in many cellular processes, including cell cycle and metabolism, both in physiological and pathological conditions. The relentless growth and proliferation of tumor cells lead to an insatiable demand for energy and nutrients, which requires the rewiring of cellular metabolism. As MYC can orchestrate all aspects of cellular metabolism, its altered regulation plays a central role in these processes, such as the Warburg effect, and is a well-established hallmark of cancer development. However, our current knowledge of MYC suggests that its spatial- and concentration-dependent contribution to tumorigenesis depends more on changes in the global or relative expression of target genes. As the direct targeting of MYC is proven to be challenging due to its relatively high toxicity, understanding its underlying regulatory mechanisms is essential for the development of tumor-selective targeted therapies. The aim of this review is to comprehensively summarize the diverse forms of MYC oncogenic deregulation, including DNA-, transcriptional- and post-translational level alterations, and their consequences for cellular metabolism. Furthermore, we also review the currently available and potentially attractive therapeutic options that exploit the vulnerability arising from the metabolic rearrangement of MYC-driven tumors.

## 1. Introduction

The MYC is a family of regulator and proto-oncogenes coding bHLHZip-type transcription factors (TFs) that have a broad range of effects on cellular processes. These include ribosome biogenesis, cell cycle, apoptosis, metabolism, and distant metastatic progress in malignant conditions [1,2]. The family consists of three human genes, including *c-MYC*, and the first of these, which has a sequence homology with the viral gene *v-myc*, was discovered 40 years ago [3]. In humans, the *MYC* proto-oncogene is located at the 8q24.21 locus. In normal conditions, it is considered to regulate the expression of approximately 15% of all genes through binding to specific enhancer box sequences, the E-boxes [4]. MYC regulates transcription through obligatory dimerization with another bHLHZip-type transcription factor, MAX [5,6]. Generally, it promotes gene expression via the release of blocked RNA polymerases, which is significantly influenced by the chromatin accessibility at target sites. Therefore, MYC is a selective amplifier, a common upstream TF that is dependent on cooperation with other TFs (such as MAX or MIZ-1) and cofactors (such as WDR5, INI1 and HSF1) bound to the target gene and/or to nearby enhancers to activate or repress gene expression [7,8,9,10,11,12].

## 2. Overview of MYC Regulation in the Physiological State

To understand the role of MYC in various cellular processes, such as metabolism, it is first necessary to review how regulation via MYC occurs in the physiological state. The expression control via MYC is a two-level process. The first level is transcriptional regulation, including different affinity of consensus sequences, variable chromatin accessibility, and interactions with other proteins. The second level is the indirect and direct (transcription-independent) role of MYC in translation (Figure 1) [13].

### 2.1. Regulation at the Transcriptional Level

Generally, MYC promotes the termination of transcription when RNA polymerase II is arrested. Moreover, it finely coordinates transcription elongation with cell cycle and replication [14]. In contrast to the pioneer TFs, which access closed chromatin, MYC binding to the target sites is highly dependent on pre-existing active chromatin, and usually, it occupies the promoter region containing E-box motifs of actively transcribed genes [15,16,17,18,19]. However, under certain circumstances, the silent chromatin can become accessible to MYC through cooperative binding promoted by other factors, such as Sox2, Oct4 and Klf4, using a partially unfolded DNA-binding domain [13,14]. In parallel with the open, active chromatin state, non-specific DNA binding is also required to allow MYC stabilization and positioning on target DNA motifs. Sequence-specific binding presumably results in allosteric changes that allow communication between the C-terminal bHLH-LZ- and N-terminal transactivation domains, which modulate transcriptional activity [15]. Furthermore, similar to the observations of MAX bHLH-LZ homodimers, subtle structural differences may also appear when MYC/MAX dimer binds to specific E-boxes or non-specific DNA sequences. This conformational selection can also help to distinguish cognate targets from non-specific ones [20].

In its canonical function, MYC, upon forming heterodimers with MAX through a highly conserved domain (comprising a helix–loop–helix and a leucine zipper domain), can bind to the high-affinity E-box consensus sequences (CACGTG) of the target genes and recruit co-regulators, such as GCN5, TIP60 and TRRAP, the part of a histone acetyltransferase complex [16,17]. High MYC/MAX concentrations are required for specific binding. However, MYC can target a wide range of DNA motifs (non-canonical E-box variants) with distinct affinities in a sequence-specific and dose-dependent manner, suggesting that the selective effects of MYC overexpression on tumor transcriptome are underrated, and the chromatin association of MYC is strongly affected by other factors [14,21]. The pathological upregulation of MYC protein causes the saturation of canonical binding sites, allowing the occupation of non-canonical bindings sites with a lower affinity. The activation or repression of these non-specific target genes may play a fundamental role in MYC-driven malignant transformation and mainly depend on three factors [12,14]. The first is that changes in MYC occupancy at individual promoters affect the size of the response, and MYC levels are regulated by promoter affinity. The second is that high levels of MYC have both positive and negative effects on the initiation of transcription, which is independent of its effect on elongation. The third is the complex formation with the zinc finger protein MIZ1, which is recruited by MYC as a repressor protein and can bind to its C-terminal sequence. Therefore, MIZ1 blocks MYC-driven transcriptional activation. The MYC/MIZ1 ratio correlates with the direction of the response [12,14].

Due to the involvement of MYC in numerous cellular processes, the tight regulation of its expression and activity at transcriptional, post-transcriptional, and post-translational levels is critical for normal function [1]. Since most of what is known about the normal physiological regulation of MYC is based on the study of tumors, a detailed discussion of MYC expression control in the context of tumorigenesis is described later in a separate chapter. In this section, we briefly describe its main regulatory mechanisms.

At the transcriptional level, the cell-type-specific enhancer regulation of MYC expression is the focus of current research (Figure 1A). For example, in both normal hematopoietic and leukemic stem cells, a super-enhancer (a cluster of multiple enhancers) located 1.7 Mb downstream of the MYC gene plays an essential role in the precise control of MYC expression [22]. Within this cluster, the individual enhancer modules selectively recruit various TFs, and they act combinatorically and additively to regulate MYC levels.

### 2.2. Indirect and Direct Translation Regulation

MYC is a modulator of ribosome biogenesis involving multiple well-coordinated steps. This is a comprehensive regulation involving the synthesis of ribosomal RNAs and proteins through chromatin remodeling and co-factor recruitment by MYC, as well as auxiliary factors (enzymes, transporters) required for ribosome assembly [23]. It is also known that the ribosomal protein RPL11 competes with the MYC coregulator TRRAP (Transformation/Transcription Domain Associated Protein). TRRAP is a core component of the TIP60 and GCN5 histone acetyltransferase complexes. Its recruitment by MYC to the promoter of target genes is essential to mediate chromatin remodeling and gene expression [24]. The direct interaction of RPL11 with MYC prevents it from TRRAP binding. In this way, RPL11 reduces H4 acetylation at MYC target promoters inhibiting MYC-mediated transactivation. As RPL11 also affects MYC expression itself and vice versa, they function as a negative feedback loop (Figure 1C) [24,25,26]. In addition to ribosome biogenesis, MYC can modulate the isoform composition of RNA polymerase III, favoring the expression of the alpha isoform (POLR3G), which has a higher transcriptional potential. This results in an elevated number of Pol III transcribed genes and unfavorable outcomes in certain cancers [27]. Finally, epigenetic regulators with intrinsic kinase and histone acetyltransferase activities, such as BRD4 (bromodomain protein 4), as well as ubiquitin ligases, such as UBE3B or lncRNAs, influence MYC protein activity and stability by changing post-translational modifications (Figure 1B). For instance, the degradation of MYC is primarily stimulated by BRD4-mediated phosphorylation of threonine 58, leading to MYC ubiquitination and degradation. As a negative feedback loop, MYC can block the histone acetyltransferase ability of BRD4 to regulate its transcription [28], while UBE3B can be repressed by the direct interaction between TRIB3 and MYC [29]. Furthermore, the nearby lncRNA PVT1 can prolong the half-life of the MYC protein by altering phosphorylation at this residue in a general, non-tissue-specific manner via physical interactions with MYC [30,31].

The direct regulation of translation by MYC is transcription-independent (Figure 1C). This process includes the induction of the mRNA 5′cap-dependent translation via the direct stimulation of its methylation through the recruitment of RNA methyltransferase (RNMT), an essential event of efficient gene expression. In this form, mRNA can bind to eIF4E, and thus to the 40S ribosomal subunit [32,33,34].

Research on non-coding, regulatory RNAs, such as miRNA or lncRNA, is one of the emerging fields of research in recent decades. Regulatory RNAs are molecules that coordinate post-transcriptionally in a very fine-grained and usually complex network during many cellular processes [35]. Several miRNAs and lncRNAs have been associated with the regulation of MYC in various tumors. Several of them act directly on the MYC 3′ UTR and promoter regions, repressing the function of MYC, as in miR24 [36], Let-7 [37,38] and miR-138 [39], or promoting it, as in CCAT1 and PVT1 [31,35], while others play an indirect role in the functional regulation of MYC in both directions (Figure 1B). For example, MNX1-AS1 promotes MYC/E2F1 stability via IGF2BP1, forming a positive feedback loop [40]. miR-3622 regulates the p53-downstream gene network, causing the p53-dependent transcriptional repression of MYC and cell cycle arrest [41].

At the post-translational level, cell-type-specific protein half-life and activity, regulated by different enzymes (e.g., kinases, ubiquitin ligases, acetyltransferases) or other interacting proteins, have a significant impact on MYC function (Figure 1B). As a key signal node, many signal transduction pathways, including WNT, NOTCH, growth factors, or IL6, converge on MYC, influencing its activation and regulation of cell growth [42,43,44,45]. Checkpoint proteins, such as p53, ARF, BIM or PTEN, are also key mediators of MYC function as they inhibit its proliferation-promoting effect and ultimately cause cell cycle arrest or apoptosis upon any failure in the process (Figure 1B). The loss of these checkpoints plays an essential role in MYC-driven tumorigenesis [41,46,47,48]. Furthermore, in a normal prostate, androgen receptors (ARs) and MYC act competitively at several shared binding sites. The ratio of AR/MYC expression determines the distribution of coactivator proteins, shifting the processes towards AR- or MYC-regulated genes (Figure 1C) [49].

## 3. Oncogenic Deregulation of MYC and Its Effect on Cancer Metabolism

In non-cancerous states, when an adequate nutrient and oxygen supply is available, cells use energy to maintain normal homeostasis, for example, to maintain membrane potential, produce proteins or replace damaged or senescent organelles and macromolecules [1]. Under starvation, as in ischemia or cancer, intrinsic and non-cell autonomous adaptive mechanisms are activated first. Among other things, these adaptive mechanisms involve main energy stores, such as white fat depots (lipolysis) and liver (glycogenolysis), and are usually capable of maintaining a sufficient circulating level of basic nutrient sources (glucose, lipoproteins, and glutamine). However, severe starvation, hypoxia, or pathological processes activate a cell’s autonomous nutrient-sensing adaptive mechanisms and reprogram the metabolism to protect the cell and maintain adequate ATP levels. The main pathways in this process are mTOR, AMPK, and GCN2 [1]. Metabolic rewiring in cancer is essential for malignant transformation or cancer cell survival and is directly driven by protooncogenes such as MYC activation and/or unique pathogenic stressors such as hypoxia. Since proliferating cells need to balance the divergent catabolic and anabolic requirements to sustain cellular homeostasis, this reprogramming allows cells to meet their increased energy and nutrient demands due to relentless growth and proliferation [50,51]. Studies in neuroblastoma and B-cell lymphoma have shown that the ubiquitin-specific peptidase USP29 is a leading protein in metabolic reprogramming. It is able to directly and selectively deubiquitinate and stabilize MYC and HIF1α, and consequently induce an adaptive response of tumor cells in both normoxia and hypoxia [51]. Another driver is the signal transducer and transcription activator STAT3, which mediates oxidative phosphorylation in AML through specific MYC activation. In turn, MYC is able to upregulate the transcription of the neutral amino acid transporter SLC1A5, increasing intracellular levels of glutamine, glutathione, and several tricarboxylic acid (TCA) cycle metabolites [52].

### 3.1. General Aspects of the Metabolic Regulation by MYC

In general, MYC activation selectively increases the expression of numerous metabolic genes required for nutrient uptake and processing to produce energy (ATP) and key cellular building blocks that increase cell mass [9]. Therefore, metabolic regulation via MYC affects a wide range of pathways, including glucose, amino acid, lipid, polyamine or mitochondrial metabolism.

#### 3.1.1. Glucose Metabolism

Aerobic glycolysis is the major carbon source of cancer cells (Warburg effect), providing a variety of anabolic precursors (metabolic intermediates) and energy, accompanied by abundant lactate production and secretion. MYC can directly activate the transcription of almost all glycolytic genes, as they contain the canonical E-box sequence [51,52,53,54,55,56]. The glucose transporter SLC2A1 (GLUT1) is a key target of MYC and allows increased glucose uptake via cancer cells by enhancing its activity [1,54,55]. In addition, MYC can modulate the isoform composition of lactate dehydrogenase enzyme, promoting LDHA over LDHB, which prefers to catalyze pyruvate to lactate transformation instead of acetyl-CoA and catalyzes pyruvate–lactate conversion rather than acetyl-CoA, which is used by default as an intermediate in the tricarboxylic acid (TCA) cycle [57]. The accumulated lactate is toxic to the cell, but through the activation of monocarboxylate transporters (MCT1 and MCT2) by MYC, cancer cells can reduce its intracellular level [58]. After the uptake of glucose, hexokinases convert MYC to glucose-6-phosphate (G6P), which is the starting point not only for glycolysis but also for the pentose phosphate pathway (PPP). The rate-limiting enzyme of PPP is G6P dehydrogenase (G6PDH), which converts G6P into 6-phosphoglucono-δ-lactone. In the oxidative phase, PPP primarily produces ribulose-5-phosphate (R5P) for nucleotide synthesis and secondarily produces NADPH for redox homeostasis (directly buffering reactive oxygen species (ROS) or indirectly regenerating oxidized glutathione), reductive biosynthesis of fatty acids, etc. [1,54,55,59,60,61]. Therefore, among other functions, MYC-driven PPP activation can protect cancer cells from oxidative stress and help directly (MYC upregulation) or indirectly (p53 loss-of-function) maintain stable DNA replication [62,63].

#### 3.1.2. Amino Acid Metabolism

Amino acid metabolism is another process tightly regulated by MYC. On the one hand, some of the essential amino acids (EAA: e.g., tryptophan) cannot be produced by cells, and non-essential amino acids (non-EAA: e.g., serine or glycine) are also of extracellular origin. In the case of enzymes involved in serine biosynthesis or transport proteins, such as SLC7A5, SLC1A5 and SLC43A1, the upregulation of MYC in a positive feedback manner could rewire the whole tumor metabolism [64,65,66]. On the other hand, glutamine transporter SLC1A5 (ASCT2) and glutaminase 1 (GLS), which are responsible for glutamine–glutamate conversion, are also upregulated by MYC. As a major nitrogen and carbon donor for nucleotides, amino acids (e.g., proline), and lipid biosynthesis, glutamine is metabolized in a variety of substrates and maintains redox homeostasis upon entry into the TCA cycle [1,50,52,54,55,67,68]. In some contexts, cancer cells become dependent on glutamine; therefore, inhibiting these first steps of glutaminolysis may lead to a slowed progression [50].

#### 3.1.3. Lipid Metabolism

The lipid synthesis coordinated by SREBP transcription factors is essential for cell membrane biogenesis as structural components, but fatty acids are also important secondary signaling molecules due to their energy storage and signal transduction [54,55,69]. Their synthesis requires large amounts of NADPH, mainly from PPP and acetyl-CoA from the TCA cycle, which is produced by the conversion of the enzyme ATP-citrate lyase (ACLY) from citrate, also regulated by MYC. Since fatty acid synthesis is an oxygen-consuming process, cancer cells in hypoxic environments often upregulate the uptake of external lipids via HIF1α instead of de novo synthesis [70].

#### 3.1.4. Polyamine Metabolism

Polyamine synthesis is also among the MYC-regulated pathways. Polyamines are involved in several essential cellular processes, from protein synthesis to stabilizing the structure of chromatin and protecting against oxidative damage [54,71,72]. MYC can directly stimulate various enzymes in several tumors, such as ornithine decarboxylase (ODC), which is responsible for ornithine–putrescine transformation, as well as spermine and spermidine synthases [54]. In hepatocellular carcinoma, the promotion of increased nutrient transport via MYC indirectly contributes to the activation of polyamine biosynthesis by mTORC1 [73].

#### 3.1.5. Mitochondrial Metabolism

Mitochondrial metabolism is essential for cells to adapt to a changing environment (e.g., nutrient supply), not only because this organelle produces energy (ATP), but also because it is the center of many biosynthetic pathways, including fatty acid or amino acid synthesis [1,59,74]. MYC is one of the main activators of mitochondrial biogenesis in tumors, acting through the induction of HIF1α and FOXO3a [1,75]. Under stress conditions, such as hypoxia, MYC can upregulate isocitrate dehydrogenase 2 (IDH2) and increase the level of alpha-ketoglutarate (αKG), an intermediate of the TCA cycle and the final product of glutaminolysis. This activation finally reduces global methylation levels, as elevated αKG induces DNA hydroxylases and RNA demethylases, while IDH2 promotes the nuclear accumulation of TETs, FTO and ALKBH5 [74].

Based on the above processes, metabolic regulation via MYC is often general, but tissue-specific modulation also occurs. One of the best examples of this specificity is the androgen-induced, prostate-specific expression of PCGEM1. This lncRNA can physically interact with MYC through its MYC-binding domain and supports the recruitment of MYC to the target promoters, enhancing its transactivation activity and increasing metabolism predominantly independent of androgen hormone or receptor status [76].

### 3.2. Distinct Processes of MYC Oncogenic Deregulation

Pan-cancer analysis shows that *MYC* is one of the most frequently deregulated oncogenes in humans and is a major driver of tumorigenesis and is associated with an unfavorable prognosis. Mechanisms underlying elevated MYC expression levels typically include copy number alterations (amplification), chromosomal translocations, somatic mutations or the deregulation of tightly regulated gene expression, such as epigenetic changes, protein stabilization or the activation of upstream signaling pathways (Table 1) [1,14,77,78]. Therefore, in the following sections, we will review the rewired tumor metabolism in relation to MYC deregulation.

#### 3.2.1. Germline Genetic Variants: Single-Nucleotide Polymorphisms (SNPs)

In recent decades, several genome-wide association studies (GWAS) have revealed several tumor-associated variants in the 8q24 chromosome region that have a real/potential impact on cancer progression [78]. The vast majority of cancer-associated SNPs are located in the regulatory regions, including enhancers and lncRNAs upstream/downstream of *MYC*, while pathogenic point mutations in the *MYC* gene itself are very rare in cancer. In Burkitt’s lymphoma, a homozygous single amino acid substitution in the transactivation domain of *MYC* was found to cause a missense mutation in two-thirds of the cell lines and biopsies examined decades ago [121].

SNPs in *MYC* regulatory regions are usually tissue-specific, but some of them, such as rs6983267 within the CCAT2 lncRNA, which causes secondary structural changes, overexpression, and consequently a higher enrichment in target sites, affect different types of cancers and have both shared and tissue-specific enhancer properties [79,80,81]. In prostate cancer (PrC), the 8q24 locus contains three independent regulatory regions upstream of *MYC* that physically loop and interact with its promoter in a tissue-specific way [82,83]. These regions carry numerous tumor-associated SNPs. In region 3, the rs6983267 SNP is associated with an increased risk for both prostate and colorectal cancer [79,84]. In prostate cancer, CCAT2-driven MYC activation can specifically increase the expression of beta-galactosidase, a key enzyme in energy production that provides monosaccharides for glycolysis [85]. Meanwhile, in colorectal cancer (CRC), rs6983267 contributes to the distinct enrichment of H3K4me1 at active and primed enhancers, p300 histone acetyltransferase and RNA polymerase II, causing chromatin remodeling and the elevated transcription of several genes [81]. rs6983267 is located within the consensus binding site of β-catenin/TCF4 transcription factors, and the G allele variant causes a stronger binding of the complex, as has been shown for the TCF7L2 factor, promoting MYC-driven CRC proliferation [79,81,86]. Furthermore, it has also been shown that rs6983267 in CCAT2 can promote an allelic-specific metabolic reprogramming of glutamine metabolism through the interaction between the G-allele and cleavage factor I (CFIm). The upregulation of the glutaminase enzyme accelerates glutamine metabolism through promoted alternative splicing, thereby promoting proliferation and metastasis [80]. The presence of rs13281615 upstream of *MYC* is associated with a higher risk of breast cancer (BC). In vitro experiments suggest that the SNP is included in a region that can physically interact with the *MYC* promoter exclusively in cancer cells but not in normal epithelial cells [84,87]. Finally, in glioma, the rs55705857 SNP in the intronic region of the CCDC26 gene adjacent to *MYC* is not only associated with IDH-mutated gliomas but also appears to confer enhanced MYC activity, potentially affecting its transactivation ability [88,89].

#### 3.2.2. Amplification

Pan-cancer analyses have shown that *MYC* is the 3rd most frequently amplified gene in malignant neoplasms, and its alterations are mutually exclusive with other genetic drivers like *PIK3CA*, *PTEN*, *APC* or *BRAF* [1,122]. Similar to *AR* in prostate cancer, *MYC* is predominantly co-amplified with the adjacent regulatory region and/or the regulatory region is amplified alone [123]. This co-selection of enhancers and oncogenes should have a selective advantage to the tumor and/or be necessary by default for oncogenic selection [124]. In pancreatic cancer, approximately 30% of tumors carry a de novo acquired, pancreatic-cancer-specific super-enhancer with marked H3K27 acetylation located near the 3′ region of *MYC* [90,91]. Furthermore, focal amplification of a 23 kb lung adenocarcinoma-specific super-enhancer ~450 kb downstream of the *MYC* promoter and a 10 kb ovarian-cancer-specific super-enhancer ~800 kb downstream of the *MYC* promoter was also detected [92].

Although gene amplifications can occur both within chromosomes and in the form of extrachromosomal circular DNA (ecDNA), previously called double minutes, the amplification of the most common oncogenes, such as *EGFR*, *FGFR2*, or *MYC*, usually occurs in the form of ecDNA [97,98,99,100]. ecDNAs are independent of normal chromosomes and may contribute to resistance to targeted therapies and shorter survival by increasing expression levels of oncogenes such as EGFRvIII ecDNA in glioblastoma [98,125]. ecDNA is also commonly found in healthy human tissues and blood, usually as a result of mutations. Approximately half of ecDNAs contain genes or gene fragments with a typical size smaller than 25 kb. The occurrence of transcription from ecDNA indicates that they remain in the nucleus, possibly influencing the phenotype and transcription of full-length and/or truncated genes. The recurrence of certain ecDNAs in different individuals, such as muscular *TTN*, indicates DNA circularization hotspots. In general, gene-rich chromosomes provide more ecDNAs [126]. Although the presence of ecDNA is not uncommon under physiological conditions, it is much more prevalent and abundant in cancer, as gene amplifications are usually greater than 200 kb in size and levels are highly heterogeneous between tumor types [98,100]. Oncogenes are shown to be particularly enriched in amplified ecDNAs, which are accompanied by elevated transcript levels that are significantly higher than those of copy-number-matched linear DNA. This is mainly due to the increased chromatin accessibility and the higher frequency of transcript fusions. This shows that ecDNA amplification differs from chromosomal amplification. It contributes more efficiently to high intratumoral heterogeneity or increased chromosomal instability and accelerates tumor evolution [98,100].

Several models exist for the biogenesis of ecDNA, including the breakage–fusion–bridge (BFB) cycle, chromothripsis, episome model, and translocation–deletion–amplification (TDA) [124,127]. In brief, according to the BFB model, telomeric loss creates a dicentric anaphase bridge via the fusion of free ends. This construct can extend during replication and be randomly fragmented under stress. Fragments then loop out via chromothripsis or another BFB cycle occurs [128,129]. Based on the chromothripsis model, subsequent chromosome shattering events, such as ss- or dsDNA breaks, some of the oncogene and/or regulatory-element-containing fragments escape from the repair mechanisms, ligate randomly, and circularize into ecDNA. In addition to the involvement of their own regulatory regions, this structural rearrangement of amplicons may be accompanied by the incorporation of new atypical active enhancers that provide physical proximity to the interaction [124,129,130]. The episome model is based on the looping out and self-expansion of the replication bubble during DNA synthesis, which may contain other DNA fragments, including transposable elements or regulatory sequences. *MYC*-containing ecDNAs stimulated by excision and amplification were previously found in leukemia [101]. Finally, the TDA model shows that the ecDNA-forming DNA fragments are produced near the breakpoints during the repair of exogenous stimuli-induced translocations. This mechanism is supported by findings in the neuroblastoma cell line SJNB-12, where the co-amplification of *MYC* and *ATBF1* occurs in the form of ecDNA after reciprocal translocation t(8;16) [102]. Any of the above mechanisms may be involved, and ecDNA serves as a multifunctional molecule in biological processes, such as epigenetic remodeling or the regulation of signal transduction, contributing to disturbed homeostasis of human cancers [127].

The complexity of ecDNAs is further enhanced by the fact that they can form clusters of 10–100 molecules in the nucleus, allowing the tumor cell to enhance expression via intermolecular interactions between oncogenes and atypic enhancers [103]. In an *MYC*-amplified CRC cell line, the BET (bromodomain and extra terminal domain) family member, a BRD4-bound *PVT1* promoter, ectopically fuses to *MYC* and duplicates, allowing the transactivation of MYC expression. However, these hubs are sensitive to the BET inhibitor JQ1. Breaking up this fusion can specifically inhibit the expression of MYC [103].

#### 3.2.3. Translocation

Chromosomal translocation is a frequent cause of oncogenic deregulation, particularly in hematopoietic cancers. The pairing of *MYC* with a highly active enhancer can separate MYC expression from normal external stimuli. For example, translocation of *MYC* and the heavy chain loci of immunoglobulins (*IGH*) was observed in about 85% of Burkitt lymphomas. This arrangement caused elevated levels of MYC expression acting through the AL928768.3 non-coding enhancer RNA and consequently promoted proliferation and therapeutic resistance [104]. In another case, in AML (acute myeloid leukemia) carrying the t(3;8)(q26;q24) translocation, the *MYC* super-enhancer region, which contains several CTCF binding sites, was relocated to the *EVI1* locus and its expression was upregulated [105].

#### 3.2.4. Deregulation of Gene Expression

The deregulation of the tightly modulated MYC expression and activity is a very complex and diverse process. MYC oncogenic activation alone is usually not enough to induce neoplastic transformation. It has long been known that the loss of checkpoint proteins, such as p53 and PTEN, in parallel with MYC deregulation, contributes to the inability of the tumor cells to switch off MYC-driven metabolism and enforces cell expansion independent of growth factors and nutrient availability [46,47,48,131,132]. Similarly, since signaling pathways converge in MYC, for example, the overactivation of growth factor or mTOR signaling causes the upregulation of MYC and its modulated processes [42,43,44,45,131]. An interesting phenomenon is that MYC overexpression is not always necessary to achieve the “MYC effect”. Similar to other malignancies, a high-fat diet and obesity are major environmental factors in the development of prostate cancer. It was shown that altered metabolism driven by dietary saturated fat intake can mimic MYC overexpression through the activation of MYC-regulated genes via hypomethylation of the H4 histone promoter at residue K20 [133].

##### Transcriptional Regulation

In addition to the above genomic alterations, MYC regulation also depends on indirect mechanisms driven by non-coding regulatory RNAs, such as lncRNA and miRNA. Because of their importance in the proper functioning of essential cellular processes, they have been an emerging area of research in recent decades. Nearby lncRNAs act as cis-regulators of MYC, influencing its effect on cellular activity, but MYC itself also modulates their expression. Therefore, the resulting feedback loops allow MYC regulation to be fine-tuned in a spatial and concentration-dependent manner [30]. Several micro RNAs have been proven to regulate MYC activity. In the physiological state, miR-24 is upregulated during the terminal differentiation of numerous lineages and inhibits the cell cycle via the direct downregulation of about 250 mRNAs. Of these, miR-24 suppresses proliferation by reducing MYC protein levels through binding to a highly complementary sequence in the 3′ UTR region [36]. The frequent downregulation of miR-24 was detected in CRC and nasopharyngeal carcinoma (NPC) [106,107]. Another important miR is let-7, which downregulates MYC in an interdependent manner. On the one hand, the RNA-binding protein (RBP), Human antigen R (HuR), binds to the 3′ UTR of MYC adjacent to the let-7 binding site and recruits let-7-loaded RISC complexes to this region [37]. On the other hand, the RBP LIN28, a suppressor of miRNA biogenesis can induce MYC expression through the inhibition of let-7 activation. LIN28 is a direct substrate of OTUD6B deubiquitinase. OTUD6B targets destabilized LIN28 in multiple myeloma, enabling the activation of let-7 [38]. In some cases, deregulation of MYC occurs at normal repressor miRNA levels and without genetic aberrations, suggesting a lack of miRNA–oncogene interactions. In CRC and hepatocellular carcinoma (HCC), the truncation of the 3′ UTR of *MYC* caused a loss of the repressor miR-138-binding site, which contributed to oncogene activation [39]. In gastric cancer (GC), since miR-3648 represses the WNT/β-catenin pathway via the inhibition of GSK-3-binding FRAT1/2 proto-oncogenes, its downregulation resulted in the upregulation of the WNT/β-catenin pathway, followed by the overexpression of MYC as a downstream effector. MYC then further downregulated miR-3648 expression by directly binding to its promoter [45]. In parallel, MYC can transactivate miR-210 and its hosting MIR210HG lncRNA, which can synergistically promote the distant metastatic process of GCs [108]. The overrepresentation of miR-3622b-3p in PrCs contributed to oncogenic processes through suppressing p53 signaling by targeting its downstream network directly (AIFM2) or indirectly (e.g., MYC) in both AR-dependent and -independent manners. This deregulation caused, among other effects, the inhibition of p53-dependent transcriptional repression of MYC [41]. Finally, in B-cell lymphoma and PrC, MYC enhanced the translation of mitochondrial GLS via miR-23a/b inhibition. This led to elevated glutaminolysis, consequently regulating energy and ROS homeostasis [109].

In addition to miRNAs, several lncRNAs are also involved in the MYC-driven tumor progression. The two most prominent and most researched lncRNAs, *CCAT1* (alias CASC19) and *PVT1* are located on opposite sides, in close vicinity to *MYC*. The upregulation of both lncRNAs was characteristic for several tumor types, including breast, gastric, ovarian, lung, esophageal, liver or colon cancers [31,35,110]. It was also shown that *MYC* amplification alone failed to increase protein levels, but co-amplification with *PVT1* may stabilize the MYC protein by protecting it from phosphorylation and subsequent degradation, suggesting that co-amplification is an obligatory event in MYC-driven tumors [31,35,111]. In vitro studies in CRC also showed that PVT1 has multiple spliced transcripts with distinct preferences for nuclear and cytoplasmic regions, assuming different roles in regulatory processes. While its circular form is mainly located in the cytoplasm (similar to MYC), most linear forms are found in the nucleus [112]. Furthermore, the *PVT1* promoter, located only 55 kb away from the *MYC* promoter, functions as an onco-suppressor in breast cancer independently of the oncogenic PVT1 lncRNA. By competing for the nearby enhancer region, the *PVT1* promoter could regulate the silencing of MYC transcription [113]. MYC-driven activation of ELFN1-AS1 was observed in early-stage CRCs. The upregulation of this lncRNA causes silencing of the tumor suppressor TPM1 by recruiting EZH2 and FOXP1 to its promoter [114,115].

Notably, cell-type-specific super-enhancers of *MYC* are particularly important as transcriptional master regulators, which are mainly located in nearby gene deserts. Their alterations, extensively discussed in previous sections, are considered to be the main mechanism of MYC deregulation [78]. Distal enhancers also contribute to the long-term transcriptional regulation of MYC. In T-cell acute lymphoblastic leukemia (T-ALL), a highly conserved NOTCH-dependent distal enhancer located at 1.47 Mb 3′ of *MYC* is frequently duplicated. Upon the binding of NOTCH1, the transcription complex physically interacts with the proximal promoter of *MYC* through chromatin loops, where NOTCH regulates the repositioning of enhancers [93,94,95]. Another lineage-specific super-enhancer, located 1.7 Mb downstream of *MYC*, was also shown to play a role in acute leukemia. The binding of the chromatin remodeling complex SWI/SNF to BRD4 can maintain the occupancy of TFs, looping with and activating the *MYC* promoter [96]. The overactivation of both nearby and distal regulatory regions causes a nonlinear increase in the expression of target genes [12].

Finally, altered expression of other proteins also contributes to the indirect regulation of MYC transcription. In prostate cancer, the redistribution of coactivators in the common super-enhancer region is dependent on the ratio of AR to MYC levels. Therefore, androgen deprivation therapy resulting in low AR may promote and accelerate the development of metastatic castration-resistant PrC via the concomitant upregulation of MYC [49,116]. In pancreatic cancer, elevated levels of the MUC5AC glycoprotein contribute to the disruption of the E-cadherin and β-catenin junction, followed by the nuclear translocation of β-catenin, which ultimately increases MYC expression levels. The upregulation of MYC then causes increased glutamine uptake and nucleotide synthesis, which contributes to gemcitabine resistance [117].

##### Translational Deregulation and Protein Stability

Translational deregulation takes place in two ways: enhanced mRNA stability and decreased protein degradation. The alteration of MNX1-AS1 at 7q36, both via amplification and MYC-induced transcriptional activation, is one of the most common aberrations in several cancers, including non-small cell lung cancer (NSCLC) and CRC [40]. MNX1-AS1 directly promotes the phase separation of the well-known RBP IGF2BP1. Therefore, it can stabilize both MYC and E2F1 mRNAs by supporting their interaction with IGF2BP1. Higher MYC levels further enhance MNX1-AS1 expression as positive feedback, and increased E2F1 activity accelerates cell cycle and proliferation [40]. The upregulation of LINC00942 lncRNA in gastric cancer contributes to the higher stability of the MYC mRNA in an N6-methyladenosine-dependent manner by preventing the ubiquitous degradation of RBP MSI2 and consequently promoting chemoresistance [118]. Finally, the hypoxia-induced increase in KB-1980E6.3 lncRNA expression in breast cancer contributes to the higher stability of MYC mRNA via IGF2BP1 recruitment [119].

The inhibition of protein degradation is another way to maintain high MYC cellular levels. This is primarily mediated by the downregulation of E3 ligases, but imbalanced ubiquitination–acetylation may also play a role, as both modifications occur on lysine residues, suggesting potential interference [134]. Loss-of-function mutations of the CUL3 scaffold protein, a core member of the E3 ubiquitin ligase complex, result in increased MYC protein levels via the elongation of the protein’s half-life [120]. The USP29 peptidase promotes elevated MYC levels via the stabilization of the MYC protein. This has an indirect positive feedback effect on the USP29 expression and consequently contributes to metabolic reprogramming [51]. Upon glucose starvation (energy stress), the upregulation of GLCC1 lncRNA is often detected in CRCs. GLCC1 can directly interact with the 5′ domain of the cytoplasmic HSP90 chaperon, dissociating it from MYC. It therefore prevents the ubiquitous degradation of the MYC protein. MYC translocation to the nucleus increases LDHA transcription and accelerates glycolytic metabolism [56].

## 4. Conclusions and Potential Therapeutic Aspects

The oncogenic overexpression of the transcriptional master regulator MYC appears to promote the global transcriptional induction of active genes involved in several metabolic processes by preferentially invading nearby active super-enhancers and remodeling the epigenetic landscape via the modulation of histone acetylation and mandatory recruitment of other protein factors [18]. This process finally results in the accumulation of MYC at promoter–enhancer hubs. The removal of architectural co-factors leads to the dissipation of MYC and a reduction in the expression of genes controlled by it. It suggests that MYC alone is unable to regulate metabolism and other cellular processes. Although MYC is not involved in the dimerization of bHLHZ proteins other than MAX, there is evidence that MAX is, expanding the MYC interactome and affecting its metabolic function [5,10,15]. It is also well-known that the oncogenic upregulation of MYC leads to metabolic reprogramming and promotes the dependence of cancer cells on exogenous glucose and glutamine. At the same time, metabolic stress activates metabolic sensors, such as MondoA, which is able to reduce glucose uptake and aerobic glycolysis via induction of thioredoxin-interacting protein (TXNIP) [135,136]. TXNIP activation decreases the genome occupancy of MYC, thereby inhibiting its transactivating function [137]. However, the functional relationship between MYC and MondoA is tumor type-specific and influenced by altered MYC levels. Their interaction can be antagonistic (as in triple-negative BC (TNBC), melanoma and PrC) or cooperative (as in neuroblastoma and B-ALL) in nutrient utilization. In TNBC, upregulated MYC inhibits the MondoA-dependent activation of TXNIP to stimulate glycolysis [138], while in *BRAF*-mutant melanoma, the MYC-induced downregulation of the MondoA-TXNIP axis leads to vemurafenib resistance [139]. The oncogenic MYC activation of glutaminolysis through GLS1 suppresses TXNIP contributing to the aggressive castration-resistant phenotype in PrCs [140]. On the other hand, in B-ALL and neuroblastoma, MYC can cooperatively promote glutamine metabolism and lipogenesis with MondoA, and the overexpression of MondoA is associated with worse clinical outcomes [135,141]. Apart from the MondoA-TXNIP axis, members of the MXD family, including MXD1-4 and MNT, also have a great impact on MYC regulation. Of these, MNT is both an antagonist and promoter of MYC oncogenic activity [5,142]. Depending on the upstream signal, MNT interacts with MLX (repression) or MAX (transcription activation) and competes with MYC for the limited amount of MAX [142,143].

MYC has a large protein network and is likely to promote cancer development in a spatial and concentration-dependent manner, depending on the global or relative level of target genes [14,18]. Therefore, understanding the underlying regulatory mechanisms is crucial to develop targeted strategies for tumor-selective therapies. Since MYC is often dysregulated in many types of cancer, it is an attractive target for tumor treatments. However, direct targeting of MYC has proven challenging because MYC is so widely used that inhibition would be globally toxic to the organism [144]. Rewired metabolism leads to therapeutic vulnerability in MYC-driven cancers and provides potential drug targets. The main challenge is to identify the specific metabolic regulatory pathways upon which tumors depend, and the designed drugs have minimal effect on the normal tissue functions [50]. Antimetabolite chemotherapy like the folic acid analog methotrexate, gemcitabine or 5-fluorouracil is a standard part of today’s modern treatments. However, their therapeutic window is narrow both due to the frequent adverse reactions and the variable efficacy in different cancer types. Oncogenic mutation of genes involved in DNA repair mechanisms can sensitize tumor cells to these nucleotide inhibitors [50,51,145].

Therefore, in MYC-driven tumors, targeting MYC-associated chromatin interactions, MYC expression, effector function, specific proteasomal degradation, and the MYC regulatory RNAs or taking advantage of synthetic lethal interactions may represent potential new strategies for cancer therapy (Figure 2, Table 2) [18,131,146].

### 4.1. Targeting Upstream Features Influencing MYC Expression

#### 4.1.1. Modulating Chromatin Interactions

The combination of the BRD4 BET inhibitor JQ1 and WNT or MAPK inhibitors is an effective therapeutic option in CRC, as the combined treatment sufficiently suppresses cell proliferation via the reduction in MYC expression (Figure 2A). This effect was mainly achieved by disrupting ecDNA hubs, for example by inhibiting the ectopic fusion of *PVT1*-*MYC* [103,147]. Stabilization of the secondary helical structure, G-quadruplex could be another strategy to inhibit MYC (Figure 2A). Within the *MYC* promoter, there is a GC-rich, nuclease-hypersensitive, parallel-stranded G-quadruplex (G4)-forming region that is critical for MYC activity. This structure is the primary target of regulation with Polypurine Reverse Hoogsteen hairpins (PPRHs) and acts as a transcriptional repressor [148,149,150]. Epigenetic modifications of this region influence the interaction with the transcription factor NM23H2, which is essential for the proper regulation of human telomerase reverse transcriptase (hTERT) and apoptosis [148]. Among others, the Quindoline derivative SYUIQ-05 preferentially binds and stabilizes the G4 region within the *MYC* promoter, but is insignificant in telomeric regions. Therefore, it prevents the expression of both MYC and hTERT, leading to cell cycle arrest [151].

#### 4.1.2. Indirect Inhibition of MYC Expression

Types of 5-aminosalicylic acid (5-ASA), such as mesalazine, have long been used in the prophylaxis of CRC. It is a potent agonist of peroxisome proliferator-activated receptor (PPAR)-γ, which can promote apoptosis by downregulating MYC. Activated PPARγ signaling appears to compensate for mutant *APC*-induced MYC expression in CRCs [152,153,154,155]. The positive reciprocal regulation of FGFR2 and CD44 was found to be important in gastric cancers. However, they differentially modulate MYC transcription. While CD44 activates it in a positive feedback method, FGFR2 inhibits MYC expression. Therefore, FGFR2 kinase inhibitors may be effective in GCs driven by *FGFR2* amplification through the parallel downregulation of CD44 (Figure 2B) [156]. Since several signaling pathways converge in MYC, the targeted modulation of other membrane receptors such as WNT, NOTCH, IL6 or other growth factors (e.g., EGFR, MET) should also be considered [42,43,44,45,185,186,187]. The therapeutic role of 5′-cap–dependent translation as a potential target in human skin squamous cell carcinoma (SCC) has been raised. Phosphorylated members of the initiation complex (eIF4E, eIF4G, and eIF4A1 factors), which are responsible for 5′-cap translation, are often upregulated. Targeting eIF4E with siRNA reduced the level of eIF4G and other proteins involved in cell cycle regulation [157].

### 4.2. Targeting MYC Effector (Downstream) Features

MYC/MAX heterodimerization is an obligatory step of MYC-driven regulation. This provides an attractive target for cancer therapy, primarily in *MYC*-amplified tumors (Figure 2C). Omomyc mini protein, an inhibitor of the pan-MYC family, is able to competitively block this process, and thus facilitate the binding of MYC to the E-box sequences of target genes, inhibiting its transactivation ability [158,159,160]. Several other experimental small-molecule inhibitors targeting MYC/MAX dimers exist, such as MYCi361 or MYCi975. By disrupting dimerization, these compounds enhance the phosphorylation of MYC on the Thr58 residue and consequently increase its proteasomal degradation. They may also contribute to increased immune infiltration and PD-L1 expression, in parallel with the sensitization of cells to anti-PD1 therapy. However, improvement in tolerability is essential as they have a narrow therapeutic index [161]. The abundance of genes regulated by MYC makes the downstream effector pathways, including alternative splicing, an attractive therapeutic alternative. Recently, it has been found that expression changes of both tumor-type-specific and pan-cancer splicing factors (comprising hubs of SRSF2/3/7) modules are frequently associated with MYC activity in several cancer types. Most of them positively correlate with MYC and induce the expression of MYC-regulated spliced isoforms [162]. In contrast, the highly homologous CDC2-like kinases CLK1 and CLK4 can phosphorylate members of the SRSF1 family, resulting in the downregulation of the genes they modulate. CLK inhibitors, such as T-025, are being investigated as anticancer drugs because *MYC* amplification makes cancer cells vulnerable to them [163,164]. Histone lysine acetyltransferases, such as GCN5, play an essential and complex role in tumorigenesis. MYC-driven GCN5 expression was proven to promote G1/S phase transition and cell cycle through the regulation of E2F1, CCND1, and CCNE1. GCN5 can directly interact with E2F1, followed by the recruitment of GCN5 to the promoter region of *E2F1*, *CCND1* and *CCNE1*. Its elevated expression is frequently observed in CRC and NSCLC, but the specific inhibitor CPTH2 can suppress the GCN5 expression causing cell cycle arrest [165,166].

### 4.3. Regulatory RNAs as Putative Therapeutic Targets

Although their clinical use has still not received much attention, non-coding RNAs (lncRNAs and miRNAs) are worth discussing separately because they have multiple functions in the modulation of MYC at many levels, both directly through epigenetic, (post)transcriptional modifications and indirectly through the regulation of PP2A and other molecules previously discussed (Figure 2D) [53]. For example, the tumor suppressors, miR-22, miR-24, or let-7, are all able to disrupt MYC E-box binding through connecting to the 3′ UTR region of *MYC*, reducing its effect on target gene transcription [36,37,167]. Reducing high MYC protein levels via the downregulation of oncogenic PVT1 lncRNA may also be an attractive strategy, as it is more accessible, and specific targeting is less harmful than MYC itself. Furthermore, the sensitization of tumor cells to chemo- and radiotherapy can be an additional advantage [31,110]. PCGEM1 lncRNA is the main prostate-specific transcriptional regulator of metabolic pathways. It is a coactivator both of MYC and AR and contributes to the reprogramming of the androgen network and central metabolism in prostate cancer cells. Because of its tissue specificity, it could be a promising target for cancer therapy in the future [76].

### 4.4. Protein Degradation

Small molecules that promote MYC ubiquitination and degradation have already been examined in preclinical studies [53]. One of the main strategies is the inhibition of deacetylases (Figure 2E). Sirtuin 2 (SIRT2), an NAD-dependent deacetylase, has a wide range of functions both in the cytoplasm and nucleus. It regulates microtubule acetylation and gluconeogenesis and is responsible for global deacetylation of H4K16 and subsequent chromatin compaction [168,169]. The thiomyristoyl lysine compound (TM) was proven to be a highly potent and specific SIRT2 inhibitor in several cancer types, promoting MYC ubiquitination and degradation, among others [170]. The inhibition of individual histone deacetylases (HDAC) is also an emerging field of cancer therapy. This is more tolerable than the previous pan-HDACi strategy. The leading compound is the HDAC6 inhibitor, which acts via the hyperacetylation of MYC (K148ac), strongly inducing its proteasomal degradation via the parallel sensitization of tumor cells to ionizing radiation [53,171,172,173]. Another attractive strategy is PPA2 activation (Figure 2E). PPA2, the major serine/threonine phosphatase in mammalian cells, is considered an onco-suppressor protein. It decreases the stability of MYC via the dephosphorylation of the S62 residue followed by ubiquitination and degradation. Pharmacological activation can be approached from two directions: the use of direct small-molecule activators (SMAPs), such as FTY-720, or the inhibition of the naturally occurring and often upregulated PPA2 suppressor, phosphoprotein SET with specific inhibitors such as OP449 [53,134]. The administration of ursodeoxycholic acid (UDCA), a natural product of the gut microbiome, was also shown to be effective in CRC. It delays G1-S transition by supporting Thr58 phosphorylation and degradation of MYC, probably via the inhibition of dephosphorylases [174,175]. Finally, USP29 ubiquitinase is a potential novel target for cancer therapy as it provides an upstream hub for MYC and HIF1α. This selective coordination is important in the regulation of distinct metabolic processes that are essential for tumor cell proliferation and growth in both normoxia and hypoxia (Figure 2E) [51].

### 4.5. Synthetic Lethal Interactions

Synthetic lethality offers an alternative therapeutic strategy for the treatment of cancer by exploiting the vulnerabilities caused by MYC deregulation. (Figure 2F). Although many synthetic lethal targets of MYC have already been identified, there is an urgent need to select the targets with the best therapeutic potential. These targets are mainly part of two key groups: regulators of MYC transcription and stability. Recently, it has been shown that simultaneous inhibition of the central coordinator of the DNA damage response (DDR), checkpoint kinase 1 (CHK1), e.g., by Chekin, and inhibition of the previously discussed epigenetic regulator BRD4, e.g., by JQ1, can be an effective and specific suppressor of HCC progression in vitro [103,176,177]. Cell cycle regulators, such as CDK1 or CDK7, are also potential effective targets. The selective inhibition of CDK1 by small-molecule inhibitors, purvalanol or roscovitine, causes the suppression of the apoptosis protein BIRC5 (survivin), which is essential for the survival of MYC-driven tumor cells [178]. The highly selective covalent THZ1 (CDK7i), targeting the remote cysteine (C312) residue of CDK7, has a high sensitivity in a subset of cancers, such as T-ALL. Among other functions, it is able to restore the expression of altered MYC [179]. Another strategy can be the direct targeting of enzymes involved in cellular metabolism (such as GLS or LDHA) or biosynthesis (such as IMPDH2). The inhibition of GLS by BPTES is based on the property that tumors are generally glutamine-dependent [180]. Likewise, the MYC-driven upregulation of LDHA significantly contributes to the Warburg effect. Therefore, the inhibition of LDHA with compounds such as NCI-737 or NCI-006 can suppress oxidative glycolysis by blocking pyruvate lactate conversion [181,182]. The guanosine biosynthetic enzymes, inosine monophosphate dehydrogenases (IMPDHs), are frequently upregulated by MYC in cancers. Since IMPDHs are rate-limiting enzymes, their blocking via mizoribine effectively reduces ribosome biogenesis and GTP biosynthesis in small cell lung cancer [183,184].

In conclusion, as a key metabolic and cell cycle regulator, MYC alterations have long been the focus of cancer research. The targeted inhibition of MYC has little clinical success due to its hub protein feature. However, MYC regulation itself and the metabolic or cell-cycle processes it regulates are extremely diverse, providing a great opportunity to exploit this to develop more specific therapeutic strategies.

## Figures and Tables

**Figure 1 cells-12-01745-f001:**
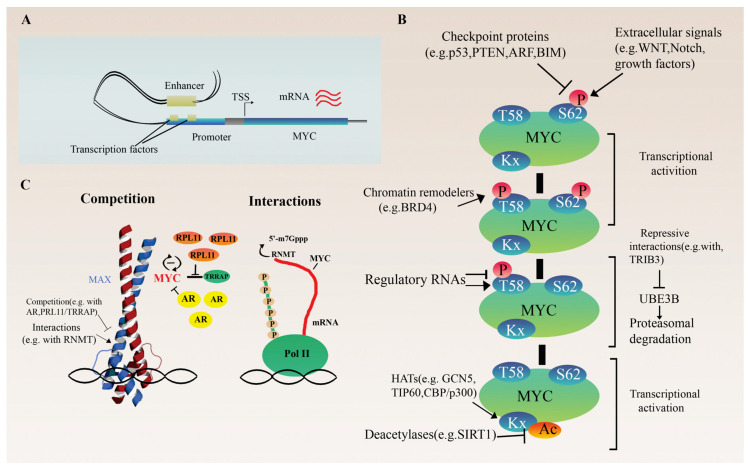
Regulation of MYC proto-oncogene. (**A**) Transcriptional regulation of MYC. The schematic illustrates the presumed looping between the super-enhancer and the promoter of an MYC gene via a cluster of cell-type-specific transcription factors and cofactors, which promote and stabilize the chromatin loop structure. (**B**) Overview of post-transcriptional and translational regulation of MYC expression. (**C**) Influencing MYC protein activity through the concentration and distribution of cofactors or direct regulation of mRNA 5′ cap methylation of target genes via interaction with RNMT.

**Figure 2 cells-12-01745-f002:**
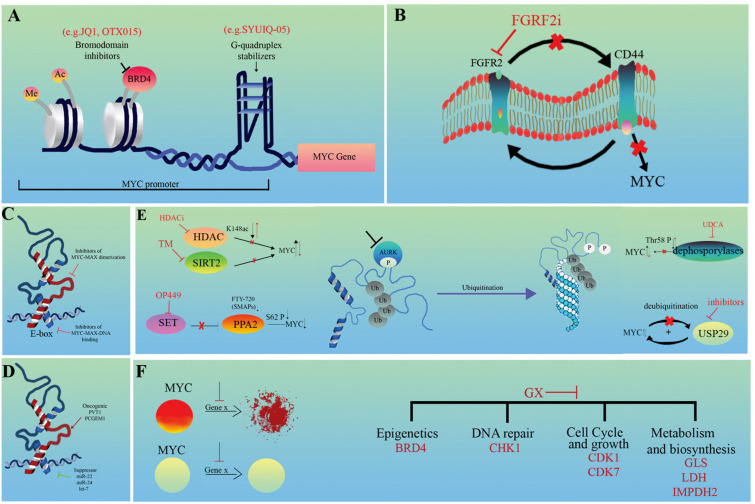
Distinct therapeutic strategies against MYC-driven tumors. (**A**) Modulation of chromatin interactions, (**B**) indirect inhibition of MYC expression, (**C**) direct inhibition of MYC/MAX heterodimerization, (**D**) indirect inhibition of MYC effector features, (**E**) targeting MYC protein degradation and (**F**) targeting MYC synthetic lethal interactions. GX represents genes targeted by drugs. Red arrows and crosses show the direction of change following the intervention.

**Table 1 cells-12-01745-t001:** Oncogenic deregulation of MYC.

Type of Alteration		Affected Molecule/Region	Cancer Types	References
Germ-line genetic variants (SNPs)		rs6983267 (*CCAT2*)	PrC, CRC	[79,80,81,82,83,84,85,86]
		rs13281615	BC	[84,87]
		rs55705857 (*CCDC26*)	Glioma	[88,89]
Super-enhancer amplification with/without *MYC*		de novo, 3′ from *MYC*	PC	[90,91]
		focal amplification, 450kb downstream	LUAD	[92]
		focal amplification, 800kb downstream	OvC	[92]
		duplication, 1.47 Mb 3′ of *MYC*	T-ALL	[93,94,95]
		focal amplification, 1.7 Mb downstream	AML	[96]
		ecDNA	multiple	[97,98,99,100,101,102,103]
Translocation		*MYC*/*IGH1*	Burkitt lymphoma	[104]
		*MYC*/*EVI1*	AML	[105]
Gene expression	Transcription	miR-24	CRC, NPC	[106,107]
		let-7	multiple	[37,38]
		miR-138	CRC, HCC	[39]
		miR-3648	GC	[45]
		miR-210 (MIR210HG)	GC	[108]
		miR-3622b-3p	PrC	[41]
		miR-23a/b	B-cell lymohoma, PrC	[109]
		CCAT1	multiple	[31,35,110]
		PVT1	multiple	[31,35,110,111,112,113]
		ELFN1-AS1	CRC	[114,115]
		Androgen receptor	PrC	[49,116]
		MUC5AC	PC	[117]
	(Post) translation	MNX1-AS1	NSCLC, CRC	[40]
		LINC00942	GC	[118]
		KB-1980E6.3	BC	[119]
		CUL3	multiple	[120]
		USP29	multiple	[51]
		GLCC1	CRC	[56]

Abbreviations: SNP—single-nucleotide polymorphism, PrC—prostate cancer, CRC—colorectal cancer, BC—breast cancer, PC—pancreatic cancer, LUAD—lung adenocarcinoma, OvC—ovarian cancer, T-ALL—T-cell acute lymphoblastic leukemia, AML—acute myeloid leukemia, ecDNA—extrachromosomal DNA, NPC—nasopharyngeal carcinoma, HCC—hepatocellular carcinoma, GC—gastric cancer, NSCLC—non-small cell lung cancer.

**Table 2 cells-12-01745-t002:** Different therapeutic aspects of MYC regulation.

	Targeted Event	Targeted Molecule/Structure	Drug (If Available)	Cancer Type	References
Targeting upstream features	Chromatin interactions	BRD4	JQ1, OTX015	multiple	[103,147]
		G-quadruplex	Quindoline derivatives (SYUIQ-05)	multiple	[148,149,150,151]
	MYC indirect inhibition	PPARγ	5-ASA (mesalazine)	CRC	[152,153,154,155]
		FGFR2	FGFR2 kinase inhibitors	GC	[156]
		eIF4E	-	SCC	[157]
Targeting downstream (effector) features		MYC/MAX heterodimerization	Omomyc, MYCi361, MYCi975	multiple	[158,159,160,161]
		CLK1/4	CLK inhibitors (T-025)	multiple	[162,163,164]
		GCN5	HAT inhibitor (CPTH2)	CRC, NSCLC	[165,166]
Regulatory RNAs		miR-22	-	multiple	[167]
		miR-24	-	CRC, NPC	[36]
		let-7	-	multiple	[37]
		PVT1	-	multiple	[31,110]
		PCGEM1	-	PrC	[76]
Protein degradation		SIRT2	TM	multiple	[168,169,170]
		HDAC	HDAC inhibitors	PC, NSCLC, BC, lymphoma	[53,171,172,173]
		PP2A	SMAPs (FTY-720)	multiple	[53,134]
		SET7	OP449	multiple	[53,133]
		dephosphorylases	UDCA	CRC	[174,175]
		USP29	-	multiple	[51]
Synthetic lethal interactions		CHK1	Chekin	HCC	[103,176,177]
		CDK1	CDK1 inhibitors (purvalanol, roscovitine)	multiple	[178]
		CDK7	CDK7 inhibitors (THZ1)	T-ALL	[179]
		GLS	GLS inhibitors (BPTES)	multiple	[180]
		LDHA	LDHA inhibitors (NCI-737, NCI-006)	multiple	[181,182]
		IMPDH2	IMPDH inhibitors (mizoribine)	SCLC	[183,184]

Abbreviations: BRD4—bromodomain 4, 5-ASA—5-aminosalicylic acid, CRC—colorectal cancer, GC—gastric cancer, SCC—skin squamous cell carcinoma, HAT—histone acetyltransferase, NSCLC—non-small cell lung cancer, NPC—nasopharyngeal cancer, PrC—prostate cancer, TM—thiomyristoyl lysine compound, PC—pancreatic cancer. BC—breast cancer, SMAP—small molecule activator, UDCA—ursodeoxycholic acid, HCC—hepatocellular carcinoma, T-ALL—T-cell acute lymphoblastic leukemia, GLS—glutaminase, LDHA—lactate dehydrogenase A, IMPDH2—inosine monophosphate dehydrogenases 2, SCLC—small cell lung cancer.

## Data Availability

Not applicable.

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
