# Peer review of "Rewired Metabolism Caused by the Oncogenic Deregulation of MYC as an Attractive Therapeutic Target in Cancers"

_cells, 2023, doi:10.3390/cells12131745_

Round 1

Reviewer 1 Report

This review presents a detailed overview of myc function in homeostatic and oncogenic settings with a focus on metabolic reprogramming. Although a lot of information is presented it would benefit from some revision to clarify and narrow the focus to the title: “Rewired metabolism….”

Suggest a first figure overviewing transcriptional, post-transcriptional/translational regulation of myc and by myc (myc targets)

In general, text could be much more succinct. There is a fair bit of repetition. Since a lot of the text simply lists data, it would be helpful to collect section 3 and section 4 lists into two tables with column 1 taken from subheadings, Column 2 collecting information relevant to subheading, column 3 with relevant cancer type(s) and a separate column for references. This would greatly enhance the usefulness and clarity of the information presented. The tables would provide an excellent resource and text for Sections 3 and 4 could then be condensed to focus on the challenges/successes/gaps of what is known, which is currently somewhat lost in the listing of information.

Minor edits:

·      Lines 99-121 should be moved to a separate section on post-transcriptional/post-translational regulation or into section on translation

·      Ln 128 : helpful to remind reader of TRAPP function here

·      Section 2.2. Need clearer distinction between direct and indirect translational regulation in the text

·      Global change (incl Ln 313, 435): change “ovarium” to “ovarian”

·      Global change: “parallelly” to “in parallel”

·      Ln 590: what is meant by “modul”?

·      Ln 609: what is meant by “basically oncogenic”? Insert ref to substantiate statement ln 608-611

Fig 1: in the file I reviewed some of the fonts were wavy and this detracted from clarity; define ecDNA in legend

Fig 2: define GX in legend

The text would benefit from attentive spelling and grammar checks.

Author Response

First of all, thank you for your valuable comments and suggestions.

  1. We carried out a comprehensive grammar and spelling check. Where necessary, we have made changes to sentences, word order, etc. without altering the content.
  2. As proposed, a new “Figure 1” was created overviewing the regulation of MYC.
  3. “In general, text could be much more succinct. There is a fair bit of repetition.”

We would like to emphasize certain things, and on the other hand, it is difficult to separate "normal" and "oncogenic" function, as most of our information comes from studies of tumors. However, we definitely wanted to talk about normal processes as well. If this is confusing, we can change the text to remove repetitions.

  1. We created Tables 1 and 2 to summarize the key information from Section 3 and Section 4, respectively.
  2. “Lines 99-121 should be moved to a separate section on post-transcriptional/post-translational regulation or into section on translation”

Indeed, it is in the wrong place. We transferred it to the regulation of translation section.

  1. “Ln 128: helpful to remind reader of TRAPP function here”

We included it in the text: “TRRAP is a core component of the TIP60 and GCN5 histone acetyltransferase complexes. Its recruitment by MYC to the promoter of target genes is essential to mediate chromatin remodeling and gene expression [Ref 40].”

  1. “Section 2.2. Need clearer distinction between direct and indirect translational regulation in the text”

We rearranged this section to be clearer. Direct regulation is transcription-independent. These MYC functions are not affected by loss-of-function mutations in the DNA binding domain as opposed to indirect regulation. Direct regulation involves the promotion of mRNA 5’ cap methylation. We put it into a separate paragraph. In addition, direct regulation of DNA replication is also a special MYC function. MYC promotes initiation by binding to several components of the pre-replicative complex and localizes them to the site of replication (promoting S-phase) [Cole M. et al. (2008) DOI: 10.1038/nrm2467]. Therefore, MYC can directly affect genome stability.

  1. “Ln 590: what is meant by “modul”?”

CLK1 and CLK4 isoforms (located at different chromosomes) are almost identical in amino acid sequence, and both are involved in the modulation of stress-induced mRNA splicing. Modul here refers to their similar function (regulating a similar set of splicing factors) and co-regulation by small molecules. We changed this sentence to be more understandable: “In contrast, the highly homologous CDC2-like kinases CLK1 and CLK4 can phosphorylate the members of the SRSF1 family, resulting in downregulation of genes they modulate.”

  1. “Ln 609: what is meant by “basically oncogenic”? Insert ref to substantiate statement ln 608-611”

The "basically" here refers to the fact that PVT1 has multiple splice variants and the transcript variant composition changes during the malignant transformation compared to normal. This change in the composition also affects its function. Since we just talk about its oncogenic function, it perhaps is not necessary in this context, so this word has been removed from the sentence.

The omitted reference has been inserted at the same time (ref #121 - Zamani, M., et al., CASC11 and PVT1 spliced transcripts play an oncogenic role in colorectal carcinogenesis. Front Oncol, 2022. 12: p. 954634.).

  1. “Fig 1: in the file I reviewed some of the fonts were wavy and this detracted from clarity; define ecDNA in legend”

This figure became a graphical abstract.

  1. “Fig 2: define GX in legend”

We defined GX in the figure legend.

Reviewer 2 Report

This is a well structured and nearly comprehensive review covering regulation and down stream effects of MYC in physiological states and oncogenic deregulation and their potential therapeutic targeting.

The sole major concern of this reviewer is the scarce discussion of the complexity of Myc interactom beyond MAX, i.e. interaction of MYC and MAX with MNT; MLX, MLSXIP and resulting important aspects of metabolic fine tuning not covering work by Ayre others e.g. (but not limited to) PMID: 2438481, PMID: 36930677, PMID: 33908607, PMID: 30103944, PMID: 28230739, PMID: 26469830, PMID: 25870263, PMID: 25640402  ...

Minor: Antimetabolite (in contrast to alkylating agents) chemotherapy is NOT  genotoxic (mainly on normal cells with high proliferation rate)

There is some awkward wording or grammar errors e.g. in lines:

75 (MYC is strongly affect by other factors), 

80 (changes in the MYC occupancy affects the size) 

104 - 105 :promoter region repressing such as miR24 [24], Let-7 [25, 26] or miR-138 [27] or promoting such as CCAT1, PVT1 [23, 28] MYC functions. Whereas others have an indirect role in the

110: At posttranslational level, cell-type specific protein half-life and activity control by 

124: It means a comprehensive regulation involving the synthesis of ribosomal 

167 - 168:  It can directly and selectively deubiquitinates and stabilizes MYC and HIF1α and consequently triggers ...

There is some awkward wording or grammar errors e.g. in lines:

75 (MYC is strongly affect by other factors), 

80 (changes in the MYC occupancy affects the size) 

104 - 105 :promoter region repressing such as miR24 [24], Let-7 [25, 26] or miR-138 [27] or promoting such as CCAT1, PVT1 [23, 28] MYC functions. Whereas others have an indirect role in the

110: At posttranslational level, cell-type specific protein half-life and activity control by 

124: It means a comprehensive regulation involving the synthesis of ribosomal 

167 - 168:  It can directly and selectively deubiquitinates and stabilizes MYC and HIF1α and consequently triggers ...

Author Response

First of all, thank you for your valuable comments and suggestions.

  1. We carried out a comprehensive grammar and spelling check. Where necessary, we have made changes to sentences, word order, etc. without altering the content.

Modifications of the highlighted sentences are also included in our answer:

“104 - 105: promoter region repressing such as miR24, Let-7 or miR-138 or promoting such as CCAT1, PVT1 MYC functions. Whereas others have an indirect role in the”

It was corrected to “Several of them act directly on the MYC 3’ UTR and promoter regions, repressing the function of MYC, such as miR24, Let-7 and miR-138, or promoting it, such as CCAT1 and PVT1, while others have an indirect role in the functional regulation of MYC in both directions.”

“110: At posttranslational level, cell-type specific protein half-life and activity control by”

It was corrected to “At the post-translational level, cell type-specific protein half-life and activity, regulated by different enzymes (e.g., kinases, ubiquitin ligases, acetyltransferases) or other interacting proteins, have a significant impact on MYC function.”

“124: It means a comprehensive regulation involving the synthesis of ribosomal”

It was corrected to “This is a comprehensive regulation involving the synthesis of ribosomal RNAs and proteins through chromatin remodeling and co-factor recruitment by MYC, as well as auxiliary factors (enzymes, transporters) required for ribosome assembly.”

“167 - 168:  It can directly and selectively deubiquitinates and stabilizes MYC and HIF1α and consequently triggers ...”

It was corrected to “It is able to directly and selectively deubiquitinate and stabilize MYC and HIF1α, and consequently induce an adaptive response of tumor cells in both normoxia and hypoxia.”

  1. “The sole major concern of this reviewer is the scarce discussion of the complexity of Myc interactom beyond MAX, i.e. interaction of MYC and MAX with MNT; MLX, MLSXIP and resulting important aspects of metabolic fine tuning not covering work by Ayre others e.g. (but not limited to) PMID: 2438481, PMID: 36930677, PMID: 33908607, PMID: 30103944, PMID: 28230739, PMID: 26469830, PMID: 25870263, PMID: 25640402…”

A new paragraph was inserted into the Discussion about the expanded MYC interactome.

  1. “Antimetabolite (in contrast to alkylating agents) chemotherapy is NOT genotoxic (mainly on normal cells with high proliferation rate)”

It was corrected (removed).

Round 2

Reviewer 1 Report

Comments have been adequately addressed. The new tables are a good addition.

Some minor points remain:

Fig 1B: “wit” > wnt

Fig 1 C: This figure is not clear. It is called out in the text as showing a negative feedback loop between RPL11 and myc but this is not apparent in the figure. In the figure legend it refers to 5’ CAP methylation. Labelling of the structures represented in the figure would be helpful.

Table 1: Transcription – targets, cancer type and references are misaligned

Table 2 : Synthetic lethal interactions – check alignment

Ln 1264 “lund” > lung

Author Response

Thank you for the suggestion for further refinments, we made the following modifications according to the reviewer suggestion.

Figure 1 part C was extended according to the reviewer`s suggestion.
Figure 1B - the typo was corrected: instead of "wit" the correct word is "with".
Tables are reformatted.  We checked the references in the tables, but they are matched with the content. The page orientation was the possible reason for the miss alignment.
In line 1264 we corrected the typo: instead of “lund” the correct word is "lung".

Reviewer 2 Report

All concerns have been appropriately met now.

Author Response

Thank you for the review!